# Second Victim Support at the Core of Severe Adverse Event Investigation

**DOI:** 10.3390/ijerph192416850

**Published:** 2022-12-15

**Authors:** Angel Cobos-Vargas, Pastora Pérez-Pérez, María Núñez-Núñez, Eloísa Casado-Fernández, Aurora Bueno-Cavanillas

**Affiliations:** 1Intensive Care Department, Clínico San Cecilio University Hospital, 18016 Granada, Spain; 2Patient Safety Committee, Clínico San Cecilio University Hospital, 18016 Granada, Spain; 3Territorial Unit II, Provincia San Juan de Dios de España, 41005 Seville, Spain; 4Biosanitary Research Institute, Ibs.Granada, 18012 Granada, Spain; 5Pharmacy Department, Clínico San Cecilio University Hospital, 18016 Granada, Spain; 6Consortium for Biomedical Research in Epidemiology and Public Health (CIBERESP-Spain), 28029 Madrid, Spain; 7Clinical Documentation Unit, Clínico San Cecilio University Hospital, 18016 Granada, Spain; 8Department of Preventive Medicine and Public Health, University of Granada, 18016 Granada, Spain

**Keywords:** second victim, adverse events, patient safety

## Abstract

There is limited evidence and a lack of standard operating procedures to address the impact of serious adverse events (SAE) on healthcare workers. We aimed to share two years’ experience of a second victim support intervention integrated into the SAE management program conducted in a 500-bed University Hospital in Granada, Spain. The intervention strategy, based on the “forYOU” model, was structured into three levels of support according to the degree of affliction and the emotional needs of the professionals. A semi-structured survey of all workers involved in an SAE was used to identify potential second victims. Between 2020 and 2021, the SAE operating procedure was activated 23 times. All healthcare workers involved in an SAE (*n* = 135) received second-level support. The majority were physicians (51.2%), followed by nurses (26.7%). Only 58 (43.0%) received first-level emotional support and 47 (34.8%) met “second victim” criteria. Seven workers (14.9%) required third-level support. A progressive increase in the notification rates was observed. Acceptance of the procedure by professionals and managers was high. This novel approach improved the number of workers reached by the trained staff; promoted the visibility of actions taken during SAE management and helped foster patient safety culture in our setting.

## 1. Introduction

The World Health Organization (WHO) recognizes the existence of a certain degree of danger, inherent in every element of the healthcare process. Safety incidents may cause unnecessary and unintentional harm to patients. When these safety breaks result in death or a life-threatening event for the patient it is considered a serious adverse event (SAE) or sentinel event [1]. These events may be used as triggers by institutions to identify potential safety issues and establish protocols to prevent them [2,3].

“Many errors are built into existing routines and devices, setting up the unwitting physician and patient for disaster. Additionally, although patients are the first and obvious victims of medical mistakes, doctors are wounded by the same errors: they are the second victims”. This is how Dr. Wu described for the first time the term “second victim” in 2000 [4]. In 2009, Scott et al. introduced a more detailed definition of second victims: “A health care provider involved in an unanticipated adverse patient event, medical error, and/or a patient-related injury who become victimized in the sense that the provider is traumatized by the event. Frequently second victims feel personally responsible for the unexpected patient outcomes and feel as though they have failed their patient, second guessing their clinical skills and knowledge base” [5].

From this concept, several studies have shown that these professionals present a picture compatible with post-traumatic stress syndrome: feelings of guilt, anxiety, affective and depressive symptoms, morbid concern about their performance and professional capacity, which can affect their clinical decision-making, and fear of legal consequences and loss of professional reputation [6,7,8]. Although it is difficult to accurately estimate the frequency of adverse events, about 15% of healthcare professionals are considered to be involved in an SAE per year [9]. The prevalence of healthcare second victims in health-care settings reported ranges from 10.4% to 43.3% [10,11]. Recent studies have shown higher prevalence, e.g., the studies conducted in Spain and Germany have shown that around 60% of physicians reported having experienced “the second victim phenomenon” at least once during their working lives [7,12,13]. To this must be added the impact of adverse events on healthcare institutions and organizations [14].

The impact of an unanticipated event is broad, affecting colleagues and future patients beyond individual harm. The creation of multi-faceted coordinated institutional support systems is important. Those systems should focus on improving safety culture, and developing and establishing contingency plans including second victim support programs, open and transparent communication with patients and families, and a communication plan to help protect the institutional image [14,15]. Several support programs for healthcare workers involved in medical adverse events have been reported with clear positive effects. They all aim to reduce emotional distress, foster coping strategies, and promote individual resilience and professional improvement [16,17]. One of the most extended support programs was developed by the University of Missouri Health Care, called “forYOU” Team. Their model is based on immediate attention to the affected workers, both emotionally and professionally, and the level of support provided to workers is stratified into three levels according to their specific needs after evaluation by a formally trained operational team [5,18].

This study aimed to share the second victim support strategy integrated into a sentinel event operational procedure developed and implemented in a university hospital in southern Spain, and to describe its accumulated results over two years.

## 2. Materials and Methods

### 2.1. Design and Site

We conducted a descriptive study of the second victim support strategy included in the operational Procedure for Serious Adverse Events (PSAE), developed and implemented in a university hospital in Granada, Spain from 2020 to 2021. The Clinico San Cecilio University Hospital is a public 500-bed hospital with over 3500 employees and a 500,000 referral urban and rural population. During the study period, the median number of inpatient admissions per year was 21,160, hospital days were 147,092 per year and the average length of stay of 6.84 days [19].

### 2.2. Participants

De-identified reports from all healthcare workers involved in an SAE where the PSAE was activated during the study period were included. We have included as an SAE any serious clinical incidents that have caused or could have caused serious harm or death of a patient, but also those with a huge impact on professionals, whatever the patient harm was.

### 2.3. Procedure

The procedure integrates care for patients and relatives, healthcare workers involved and the institution, as well as the investigation of the factors associated with the event and the development of improvement actions. It consists of different phases, starting with the identification of the SAE and ending with the evaluation of the actions included in the improvement plan and the recovery of the professionals. A flow chart of the procedure is shown in Figure 1.

A description of the different phases of the procedure is detailed in Appendix A. For the management of the adverse event, a two phase semi-structured interview of every involved worker is conducted; firstly, an evaluation of workers involved in the event to evidence potential signs or symptoms of a second victim, and secondly, the professional’s contribution to the investigation of the event and their perception of the possible causes. Finally, needed changes or improvements to the infrastructure or organization are listed, detailing evaluation procedures and due dates.

### 2.4. Variables and Data Analysis

We collected the number of events reported and adverse events triggering the PSAE. For each serious adverse event investigated, the total healthcare professionals involved, the number and features of healthcare workers showing signs and symptoms of second victims, and the actions taken by the PSAE team was recorded. We presented here the absolute figures and the frequency distribution. Interviewed healthcare workers’ commentaries related to the procedure were also included for evaluation and improvement purposes.

## 3. Results

From January 2020 to December 2021, 447 patient safety incidents were reported to the institutional Adverse Event Reporting System [https://www.seguridadpaciente.es/sistema-de-notificaciones/, accessed on 1 November 2022]. From those, 6 (1.36%) resulted in a fatal outcome and 24 (5.44%) had a severe impact on the patient; 74 (16.55%) had a moderate impact and 132 (29.48%) had a mild impact. The remaining declared events, 211 (47.17%), did not harm the patient.

The PSAE was activated 25 times, although, after a careful investigation, the steering committee deactivated two of the procedures, considering that they did not meet the criteria for an SAE. A total of 135 workers were involved in the 23 SAE investigated. Data from 10 professionals were missing (See Table 1).

The time to activation of the crisis committee ranged from 12 h to one week. On twelve occasions (48%) it was activated within 24 h, seven times (28%) activation took place between 24 and 48 h, and the remaining times activation took more than 48 h. In four out of thirteen occasions (30.76%) where the activation time was delayed, the event had occurred during the weekend, when there were no available staff from the steering committee.

Due the characteristics of the PSAE, all healthcare workers involved in an SAE (*n* = 135) received second-level support. Of those, only 58 (42.96%) reported having received emotional first aid at the first level of care. Globally, 47 (34.8%) met the criteria of “second victim”. The distribution by sex and professional category of the workers interviewed are shown in Table 2. First-level support was more frequently reported by women. No major differences were found according to professional categories. The mean number of professionals interviewed for each event was 5.9, with a minimum of 1 and a maximum of 17, and in only three cases was only one professional involved.

A description of workers identified as second victims by the second-level team and the level of support needed for their recovery is detailed in Table 3. Overall, the majority of second victims self-reported that they were able to discuss the issue with their coworkers (*n* = 26), 10 with friends or relatives (*n* = 10), while 11 (23.4%) reported that they were not able to talk to anyone. Only two out of forty-seven (4.26%) needed to take a sick leave and another one refused to go on working 24 h shifts after the event. A total of four reported guilt feelings, and six workers suffered from sleep disturbances that required pharmacological treatment. Among those forty-seven professionals with signs or symptoms of second victims, seven (14.9%) workers had solved their problem exclusively with the first emotional help provided by the first level of support, and it was noteworthy that the higher percentage of response to first level emotional support was among intern physicians. Referral to the second and third level of care support was indicated in thirty-three (70.2%) and seven (14.9%) cases, respectively. Four out of seven professionals (57.1%) refused the specialized support offered by the third level, so they were followed up by telephone until their recovery by the second-level team. Among all workers involved in an SAE, thirty-three (24.44%) acknowledged that they would have appreciated being able to leave the hospital after the event occurrence.

From a qualitative point of view, the following reflections of the professionals involved can be highlighted and may help to identify the strengths and weaknesses of the procedure. In the words of the professionals themselves:“The procedure must be known to all professionals”.“Supervisors and Heads of units need to be better trained and more aware”. “There is a problem with the first- level approach”.“We are afraid and uncertain about the consequences when an adverse event occurs. This kind of project helps to solve this”.“After the experience with this procedure, we know the hospital also takes care of the professionals. This is reassuring for us”.“The interview conducted at level two is really important and what we value the most. We feel supported and listened to”.“Being able to participate in the improvement actions has been very important for us”. “We feel that we are not alone”.

## 4. Discussion

We described the two years’ experience of a second victim support intervention developed from the strategy designed by the “forYOU” program built on the Scott Three-Tiered Interventional Model of Support for Second Victims [18] and integrated into the operational PSAE. The main finding resulting from the interviews conducted within the PSAE was that one in three healthcare workers involved in an SAE was highly affected both emotionally and professionally and showed signs or symptoms of a “second victim”. Workers reported insufficient frequency of first-level emotional support. This level of care provided by peer colleagues helped workers to recover from the experience in less than one out of five cases. The fact that the second victim support intervention was at the core of the SAE investigation helped the institution to reach out to at-risk workers. The qualitative approach showed that the assessment of the procedure among the interviewed professionals was very positive, although of most of them deemed the procedure to be little known. A secondary benefit, not shown in the previous analysis, was the increased number of notifications to the sentinel event program compared with previous periods (over 31%). We strongly believe that this result may be influenced by a major influx in safety culture after the implementation and dissemination of the second victim strategy in our hospital.

### 4.1. Strengths and Limitations

The experience reported refers to a single, medium-sized hospital and covers a limited period of two years. Although promising, it remains to be confirmed whether the procedure can be successfully adapted to other settings. The characteristics and composition of the crisis committee are key factors to account for and consider. In our hospital, the workers leading the program are well-known, recognized as empathetic, and respected. All of them are experienced and have received specific training in adverse event management and second victim support. Despite the strong support from the hospital management team, none of them is directly involved in the interview phase of the procedure. The committee operates autonomously, both in conducting the second victim’s interview and in leading root cause analysis.

Awareness of the described procedure among hospital professionals was insufficient, in spite of its availability on the hospital website and dissemination efforts in any clinical units. Most healthcare professionals are unaware of its existence or its full content. It is necessary to enhance training and dissemination actions at every moment and at all levels. SAE management has been identified as a key opportunity to disseminate the procedure and, therefore, enhance patient safety culture.

The particularity of our procedure is that the activation of the PSAE integrates the approach of all stakeholders: patients, workers and the institution. Regarding second victim management, the main difference with respect to the model proposed by the “forYOU” team [18] is that all professionals involved in an SAE automatically receive support from properly trained peers, regardless of whether or not they have received first-level support. Other approaches, such as RISE [20], are also activated only after the affected professional’s demand. In this sense, the sensitivity expected to identify workers who need support may be enhanced, as the recruitment will not rely exclusively on the second-victim demand. Thus, the program’s access is not limited by stigma and other barriers faced by the health workers involved in an SAE [21].

Regarding the study design, we conducted a retrospective evaluation of an operational procedure conceived for day-to-day hospital activities and not for research purposes. Since preserving the confidentiality of the workers was key to the program, there were no records of the interviews conducted, and therefore a systematic evaluation (qualitative or quantitative) could be made regarding second victims’ symptoms or their reflections from the SAE. Neither information on the unit managers in charge of the first-level care support or information related to the recovery process of the workers who accepted third-level care support were included. These important aspects, as well as the analysis of the economic impact of the PSAE, would be desirable to address in future research.

### 4.2. Practical Implications

The importance of having a PSAE in place is undisputed [14,15]. In our experience, the integration of the second victim support program as another process within the investigation of an SAE was feasible, with a positive impact and, probably, easy to adapt and generalize to other settings. To ensure the success of the program, it is key that they are recognized as institutional programs. Championing leadership, independence, and commitment to patient safety, with time for these duties, is also essential.

The integration of the second-victim support program into the investigation of an SAE allows the proactive identification of workers at higher risk. Regardless of the degree of affectation of the professional, the initial attention to their emotional well-being increases their sense of belonging, increases their confidence in the institution, and facilitates their collaboration in the identification and correction of factors associated with the occurrence of the event. Only when the employees stop blaming themselves for the error or its consequences, can they effectively participate in the identification of modifiable risk factors. These strategies foster institutional safety culture by helping healthcare workers to understand and to internalize that reporting adverse events not only benefits patients, but also themselves, without compromising their prestige or their job.

Over one-third of workers involved in an SAE developed signs or symptoms of second victim (34.8%). In 14.9% of them, a third level of specialized support was needed. This percentage was higher than the 10% reported by Scott et al. [18]; however, we have only considered severe adverse events. This figure was particularly high among consultant physicians (29.4%), probably due to their greater responsibility for patient wellbeing.

The fact that only half of the professionals reported receiving first-level support according to the established procedure draws attention. The low percentage could be explained due to the already existing unstructured and informal peer support and, therefore, a formal system could be redundant, which is consistent with the findings of “The Buddy Study” [22]. On the other hand, the first level of support was sufficient in only 14.9% of second victims, a much lower percentage than the 60% reported by the “forYOU” team based on estimations [18]. As mentioned above, our procedure guarantees that properly trained peer support is offered to 100% of the professionals. It is crucial to have a team of instructed workers giving answers to uncertainties expressed by the professionals, generally related to the event investigation process and its potential consequences, including legal issues. This level met the needs of 70.2% of the second victims, more than double (30%) the estimation made by the “forYOU” Team program [18].

## 5. Conclusions

In our experience, the routine integration of the second victim support into the SAE investigation allows all workers involved in an SAE to be reached by trained peers. We have identified wide room for improvement at the first level of support provided. The proposed model of intervention is feasible, favors the systematic detection of workers affected by a patient safety break and boosts their implication in the event investigation (identification, establishment and evaluation). Placing second victims at the core of SAE management procedure helped foster patient safety culture and enhance the visibility of improvement interventions resulting from adverse events investigation.

## Figures and Tables

**Figure 1 ijerph-19-16850-f001:**
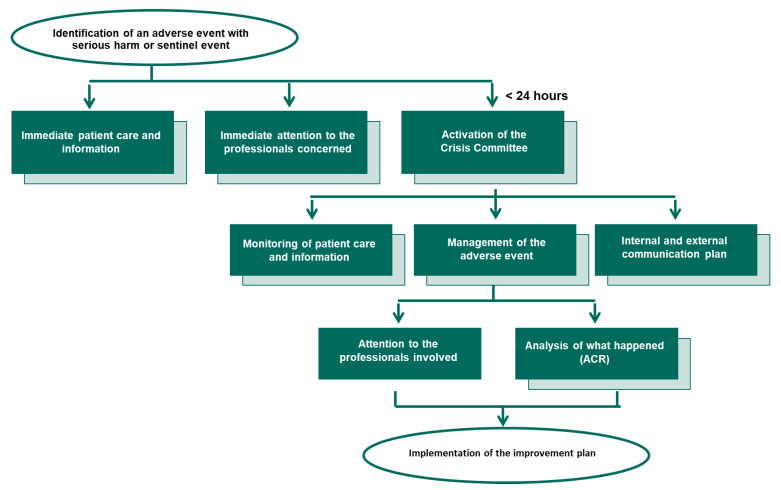
Flow chart of the operational Procedure for Serious Adverse Events.

**Table 1 ijerph-19-16850-t001:** Features of the procedure for Serious Adverse Events activated during the study period.

Adverse Event (Numerical Order)	Time to Activation (Hours *)	Clinical Unit Type Where the Adverse Event Took Place	Number of Workers Interviewed	Severity of Adverse Event	Number of Second Victims Identified
1	24	Surgical	6	Severe	3
2	24	Medical	6	Moderate–Severe	4
3	24	Diagnostic support	8	Severe	0
4	24	Medical and surgical	10	Severe	3
5	24	Medical	9	Severe	3
6	7 days	Medical	6	Severe	4
7	72	Surgical	3	Severe	0
8	48	Diagnostic support	6	Severe	3
9	48	Surgical	3	Moderate–Severe	2
10	48	Surgical	3	Severe	3
11	48	Medical	6	Severe	3
12	24	Surgical	13	Severe	2
13	24	Medical	3	Not available	1
14	72	Medical	2	Moderate–Severe	2
15	24	Emergency and Surgical	17	Severe	7
16	24	Medical	8	Severe	2
17	24	Surgical	10	Severe	2
18	100 days	Surgical	1	Severe	0
19	48	Medical	8	Severe	0
20	24	Surgical	1	Moderate–Severe	0
21	24	Surgical	2	Severe	0
22	48	Surgical	1	Severe	1
23	7 days	Surgical	3	Severe	2

* Hours except where days are specified.

**Table 2 ijerph-19-16850-t002:** Distribution by sex and professional category of the healthcare workers interviewed according to the self-reported first level of support received and those identified by the program as “second victims”.

		Health Workers Interviewed *n* (%)	Self-Reported First Level of Support Received *n* (%)	Second Victim Signs and/or Symptoms *n* (%)
Sex	Male	33 (24.4%)	12 (36.4%)	10 (30.3%)
	Female	92 (68.1%)	43 (46.7%)	35 (38.0%)
	Missing data	10 (7.4%)	-	-
Professional category	Consultant physicians	43 (31.9%)	19 (44.2%)	17 (39.5%)
	Interns/residents physicians	26 (19.3%)	11 (42.3%)	8 (30.8%)
	Nurses	36 (26.7%)	16 (44.4%)	13 (36.1%)
	Nursing assistants	15 (11.1%)	7 (46.7%)	6 (40.0%)
	Radiology technicians	4 (3.0%)	1 (25.0%)	1 (25.0%)
	Hospital porters	1 (0.8%)	1 (25.0%)	0
	Missing data	10 (7.4%)	-	-
Total		135 (100%)	58 (42,96%)	47 (34,81%)

**Table 3 ijerph-19-16850-t003:** Distribution of healthcare workers identified by the program to present second victim signs or symptoms according to the level of support needed for their recovery.

		Total	First Level of Support *n* (%)	Second Level of Support *n* (%)	Third Level of Support *n* (%)
Sex	Male	10	1 (10.0%)	7 (70.0%)	2 (20.0%)
	Female	35	6 (17.1%)	24 (68.6%)	5 (14.3%)
Professional category	Consultant physicians	17	0	12 (70.6%)	5 (29.4%)
	Interns/residents physicians	8	4 (50.0%)	4 (50.0%)	0
	Nurses	13	2 (15.4%)	9 (69.2%)	2 (15.4%)
	Nursing assistants	6	1 (16.7%)	5 (83.3%)	0
	Radiology technicians	1	0	1 (100%)	0
	Hospital porters	0	0	0	0
Total		47	7 (14.9%)	33 (70.2%)	7 (14.9%)

## Data Availability

Not applicable.

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
