# Peer review of "Second Victim Support at the Core of Severe Adverse Event Investigation"

_ijerph, 2022, doi:10.3390/ijerph192416850_

Round 1

Reviewer 1 Report

Well written article to address the impacts of SAE to HCW. Important topic. 

Can add some content to describe: 1. The symptoms and reflection of HCW suffered from SAE. 2. How about multiple involvement from more than one HCW in a SAE? 3. Details about the contents of supports, according to its level.

Author Response

See uploaded word file

Reviewer 2 Report

First of all, thank you for the opportunity to review this suggestive paper. In my opinion, I believe that the document consistently and accurately reflects the stated objective of 'share the second victim support strategy integrated into a sentinel event operational' (75-76). The flowchart of the process, clearly shows in the tables (table 1 and table 2) the nature of the adverse events and the distribution of the 'second victims'. In my opinión, is very clear and a good point of the paper.

On the other hand, I would have liked to have had more information regarding the qualitative aspect. In other words, I find that paper could improve with a more extensive description of the bases of the 'qualitative approach' (what is missing).For instance: what protocol has been followed? What type of analysis has been developed? And, similar to the quantitative results, I believe that this analysis deserves to be reflected in a more systematic way.

In any case, I think it is a relevant paper of hight interest to the healthcare community. 

Author Response

See uploaded word file
